# Endohedral Fullerene Fe@C_28_ Adsorbed on Au(111) Surface as a High-Efficiency Spin Filter: A Theoretical Study

**DOI:** 10.3390/nano9081068

**Published:** 2019-07-25

**Authors:** Ke Xu, Tie Yang, Yu Feng, Xin Ruan, Zhenyan Liu, Guijie Liang, Xiaotian Wang

**Affiliations:** 1Hubei Key Laboratory of Low Dimensional Optoelectronic Materials and Devices, Hubei University of Arts and Science, Xiangyang 441053, China; 2School of Physical Science and Technology, Southwest University, Chongqing 400715, China; 3School of Physics and Electronic Engineering, Jiangsu Normal University, Xuzhou 221116, China

**Keywords:** molecular spintronics, C_28_ endohedral fullerene, spin transport properties, spin filter

## Abstract

We present a theoretical study on the adsorption and spin transport properties of magnetic Fe@C_28_ using Ab initio calculations based on spin density functional theory and non-equilibrium Green’s function techniques. Fe@C_28_ tends to adsorb on the bridge sites in the manner of C–C bonds, and the spin-resolved transmission spectra of Fe@C_28_ molecular junctions exhibit robust transport spin polarization (TSP). Under small bias voltage, the transport properties of Fe@C_28_ are mainly determined by the spin-down channel and exhibit a large spin polarization. When compressing the right electrode, the TSP is decreased, but high spin filter efficiency (SFE) is still maintained. These theoretical results indicate that Fe@C_28_ with a large magnetic moment has potential applications in molecular spintronics.

## 1. Introduction

Molecular spintronics has attracted tremendous attentions due to its promising applications in nanoelectronics in the past few years [1,2,3,4,5,6]. The conductance can be directly controlled by the spin degrees of freedom in a single molecule. Compared with traditional semiconductor spintronic devices, the spin–orbital coupling and hyperfine interactions in the molecule are weak [7]; therefore, the molecular building blocks in spintronics are a very promising candidate for the new generation of molecular devices that can improve performance and enhance functionality, especially in magnetic storage and quantum information processing [8,9].

One of the central issues in molecular spintronics is how to manipulate the spin freedom in a molecular junction. A common approach is having the magnetic molecule sandwiched between the source and drain metallic electrodes [10]. Among all possible candidates, magnetic (endohedral) fullerenes form an attractive family of clusters due to their peculiar structures and remarkable properties. Recently, various fullerenes that include C_60_ [11,12,13,14,15], C_70_ [11,12,13,16], and B_40_ [17,18] have been extensively studied and can be suggested as magnetoresistance devices [11,13,14,15] or spin filters [16,17]. In the fullerene family, it is worth noting that the small fullerene C_28_ has inherent magnetic moment [19].

Up to now, C_28_ has been extensively studied both in terms of experimental and theoretical aspects [20,21,22,23,24,25,26,27,28]. C_28_ adopts a Td point group, with four unpaired C atoms located at the apex of a tetrahedron, which leads to large magnetic moment (4.0 *µ*_B_). However, the delocalized magnetic moment is unstable and easy to quench in C_28_ [19], which limits its application in molecular spintronics. Fortunately, encapsulated 3d or lanthanide transition metal (TM) atoms can stabilize the structure, and the magnetic moment can also be localized in the TM atoms [20,22,23,24]. Previous studies mainly focused on stability [20,21,23,24], magnetic properties [20,26,28], and vibrational properties [25]. Nevertheless, the spin transport properties of the magnetic TM@C_28_ have not been reported so far and thus require examination. Considering the stability and magnitude of the magnetic moment of 3d TM@C_28_ [20], Fe@C_28_ is very promising for molecular spintronics. Thus, two key questions should be addressed: can Fe@C_28_ be adsorbed on an Au(111) surface?; and does Fe@C_28_ junction have a high-performance for molecular spintronic devices? The answers to these questions may broaden the opportunities for designing novel fullerene devices.

In this research, the electronic structure of Fe@C_28_ was studied first. Then, we considered all possible adsorption configurations of Fe@C_28_ adsorbed on the Au(111) surface and identified the most stable configuration after structure relaxation. Finally, the spin transport properties of a scanning tunneling microscopy (STM)-type Au-Fe@C_28_-Au junction at the molecular scale were explored. The Fe@C_28_ junction can act as a high-efficiency spin filter, and the conductance can be effectively modulated by means of pushing forward or pulling back the Au tip.

## 2. Materials and Methods 

In these calculations, electronic structures and geometry relaxations were performed using the SIESTA package [29]. The exchange-correlation energy functional was described by the Perdew-Burke-Ernzerhof (PBE) generalized gradient approximation (GGA) form [30]. An energy cutoff was set to be 400 Ry for the real-space mesh size. For the relaxation of structures, the Hellmann-Feynman (HF) forces on each atom was less than 0.01 eV/Å to find a local minimum.

The spin-dependent transport properties studied are based on a state-of-the-art technique where the density functional theory (DFT) is combined with the Keldysh nonequilibrium Green’s function (NEGF) theory, implemented in the NanoDcal package [31,32]. The spin-dependent current–voltage (I-V) curves and transmission functions were obtained by using the Landauer-Büttiker formula as the following equations (1) and (2):(1)$$I_{\sigma }\left(V\right)=\frac{e}{h}\int T_{\sigma }\left(E,V\right)\left[f\left(E-\mu_{L}\right)-f\left(E-\mu_{R}\right)\right]dE$$
(2)$$T_{\sigma }\left(E,V\right)=Tr\left[\Gamma_{L}\left(E,V\right)G_{\sigma }\left(E,V\right)\Gamma_{R}\left(E,V\right)G_{\sigma }^{†}\left(E,V\right)\right]$$
Here, *I_σ_(V)* is the current through the device under the external bias voltage *V*; *T_σ_(E,V)* is the transmission function of the junction, indicating the rate at which electrons are transmitted from the left to the right electrodes by propagating through the device; *G_σ_(E,V)* is the retarded Green’s function of the central region (*σ* = ↑/↓); *Γ_L/R_(E,V)* is the left/right electrode and central region coupling matrix; *µ*_(L/R)_ is the chemical potential; and *f(E-µ_(L/R)_)* is the Fermi-Dirac distribution function. In our transport calculations, the model of the Fe@C_28_ junction was periodic along the *x* and *y* directions, and the transport direction was along *z*. A mesh cutoff energy was set to be 150 Ry, and the Monkhorst-Pack (1 × 1) K-point grid was adopted to sample the 2D Brillouin zone. Test calculations with a larger basis set, larger cutoff energy, and denser K-point (i.e., 3 × 3) provide similar results.

## 3. Results

### 3.1. Electronic Structure of Fe@C_28_

First, we will discuss the electronic properties of Fe@C_28_. The fully relaxed structure of Fe@C_28_ fullerene adopts a C2v point group (Figure 1A). The Fe atom deviates from the cage center with a 0.454-Å displacement and reduces the Td symmetry to C2v. The Fe atom is above the middle of two pentagons sharing an edge and forms two identical Fe–C bonds, with a corresponding bond length of ~2.052 Å. According to symmetry classification, there are 10 types of inequivalent C atoms, 13 inequivalent C–C bonds, and 6 inequivalent rings that include 4 inequivalent pentagons and 2 hexagons, respectively. The optimized C–C bond lengths are changed from 1.430 to 1.541 Å [20]. The binding energy of Fe@C_28_ is defined as E_Fe@C28_ – E_C28_ – E_Fe_ = −1.312 eV, indicating that the C_28_-encapsulated Fe atom makes the Fe@C_28_ much more stable. 

Figure 1B shows that the isosurface of spin density is mainly localized on Fe atoms, which is a remarkable difference in the C_28_ case [12]. The calculated magnetic moment is mainly distributed on the Fe atom, which is about 3.270 *µ*_B_, and the total magnetic moment of Fe@C_28_ is expected to be 4.0 *µ*_B_. 

In addition, we also examined the electronic structure of Fe@C_28_ with Td symmetry. By adopting the DFT calculations, the Td-symmetry Fe@C_28_ has a larger magnetic moment of 6.0 *µ*_B_, but its energy is 1.054 eV higher than that of the C2v configuration. Figure 1C–E shows the spin-resolved frontier orbitals, average density of states (DOS) of C, and Fe’s 3d orbitals of Fe@C_28_. It is obvious that there are no degenerate orbitals in Fe@C_28_, since the C2v point group only has one-dimensional representations. Clearly, the energies of spin-resolved frontier molecular orbitals is significantly differ from one another. The spin-up HOMO (3d*_xz_* dominated in Fe) and LUMO two frontier orbitals locate at −0.163 and 0.162 eV, while the spin-down HOMO and LUMO (3d*_x_*_2−*y*2_ dominated in Fe) locate at −0.571 and 0.914 eV. Then, the spin-up and spin-down electrons of the HOMO–LUMO gaps are 0.325 and 1.485 eV, respectively. Since the electronic structure of the spin-up and spin-down electrons is significantly different, Fe@C_28_ could be a potential candidate in molecular spintronics.

### 3.2. Adsorption Properties

Before the spin transport calculation, we studied the adsorption properties of Fe@C_28_ on the Au(111) surface and first identified the most stable adsorption configuration. Symmetry analysis showed that the free Fe@C_28_ with a C2v point group has a total of 29 inequivalent sites, which include 10 types of inequivalent C atoms (named point), 13 types of inequivalent C–C bonds (named line), and 6 types of inequivalent rings, respectively. Then, three special adsorption sites at the Au(111) surface were considered, namely hollow, bridge, and top sites. Therefore, there are a total of 87 adsorption configurations. To find the most stable adsorption configuration, we relaxed all the initial structures. A 4 × 4 × 3 supercell for the Au(111) surface and 11.54 × 11.54 × 30 Å^3^ were adopted to mimic the adsorption unit cell. The Monkhorst-Pack (5 × 5 × 1) K-point used to sample the Brillouin zone ensures accurate results. In this calculation, we fixed the Au substrate and only relaxed the Fe@C_28_.

The lowest energy configuration is shown in Figure 2A,B. After adsorption, C_28_ cage has a slight distortion, while the Fe atom has an obvious displacement. The Fe atom has a total displacement of 0.174 Å with respect to the undistorted Fe@C_28_, and forms a Fe–C bond of about 2.122 Å in length. The Fe@C_28_ tends to adsorb on the bridge site of the Au(111) surface via a C–C bond. The adsorbed C–C bond is nearly parallel to the Au–Au bond (Figure 2B), and the bond length is slightly elongated from 1.435 to 1.536 Å. Two Au–C bond lengths are 2.232 and 2.354 Å after adsorption, implying effective bonding between Fe@C_28_ and the Au(111) surface. We obtained the adsorption energy as follows: ΔE = E_Fe@C28+Au_sub_ – E_Fe@C28_ – E_Au_sub_ = −1.19 eV. The large adsorption energy indicates that Fe@C_28_ on Au(111) has a strong chemical adsorption.

To understand the changes in electronic structure of Fe@C_28_ before and after adsorption, a projected density of states (PDOS) analysis was performed (Figure 2C). Fe d orbital is around 1.3 eV above the Fermi level, which is similar to the free Fe@C_28_. However, due to the strong Au–C chemical bonding and charge transfer, the original sharp peaks in freestanding C_28_ are broadened and mixed together. Mülliken population analysis showed that 0.26 e are transferred from Au(111) to Fe@C_28_, and the Fe atom loses 0.16 e. On the one hand, the charge transfer makes the LUMO occupied and the peak disappear (Figure 2C). On the other hand, the magnetic moment of the Fe atom is slightly increased by 0.17 *µ*_B_. In order to see the charge transfer more clearly, we plotted the real space differential charge density distribution and average along the *z* direction, which is presented in Figure 3. It is clear that the charge transfer mainly occurs at the surface between Fe@C_28_ and the Au substrate. Furthermore, because the Fe atom is bonded with the C that is shared by the three pentagons, a significant charge transfer was also expected. In conclusion, because the magnetic moment is hardly affected by adsorption, this robust magnetic moment provides a prerequisite for molecular spintronics applications.

### 3.3. Transport Properties

We designed a STM-type junction of Fe@C_28_ and studied the voltammetric properties. The central region of the STM-type junction can be divided into two parts: the upper part is modeled as an Au adatom adsorbing on a 4 × 4 × 2 Au(111) hollow site, and the lower part is the most stable Fe@C_28_-Au(111) adsorption configuration discussed above. The central region is shown in Figure 4A. This Au-Fe@C_28_-Au junction is reasonable in experiments, and the current is easily to measure. It is well known that the contact configuration of molecules and electrodes is a key factor affecting the transport properties [33,34,35,36,37]. When pulling back and pushing forward the Au STM tip, the corresponding Au–C bond will be broken or formed. Therefore, the conductance of the junction can be modulated by controlling the distance between the Au tip and Fe@C_28_. 

The non-bonding configuration was considered first: the nearest Au–C distance was ~3.12 Å. The calculated spin-resolved I-V curves within the bias region from −1.0 to 1.0 V are shown in Figure 4B. Due to the asymmetric coupling between Fe@C_28_ and the left and right Au(111) surface, the currents under the positive and negative bias are not symmetric. We found the spin-down current to be always larger than that of the spin-up current within the examined bias range. The spin-up (down) current at ±1.0 V bias of the Fe@C_28_ junction is 8.26 (14.75) and −7.67 (−20.92) *µ*A, respectively. To determine the difference between the spin-up and spin-down current, we defined the bias voltage-dependent quantity as *R(V)* = |*I_down_(V)*/*I_up_(V)*|. The calculated R from 2.52 to 12.05 shows effective spin polarization within the examined bias range. Furthermore, the current under the negative bias voltage is significantly larger than that of the positive bias voltage, which indicates a moderate rectification. Our results suggest that the Fe@C_28_ junction is a high-efficiency spin injector under small bias voltage.

We subsequently studied the spin transport properties of the Fe@C_28_ junction in equilibrium. The spin-resolved transmission spectra of the Fe@C_28_ junction is plotted in Figure 5A, which shows a significant difference between the spin-up and spin-down channels nearby the Fermi level. The transmission spectra do not behave as several discrete and sharp transmission peaks but as a continuous broadening platform-like peaks for both the up and down channels, which indicates a strong molecule–electrodes coupling. For the spin-up channel, there are two 0.5-eV width platforms of transmission spectra that are located at regions from −1.0 to −0.5 eV and from 0.5 to 1.0 eV, respectively. As the transmission coefficients tend to zero nearby the Fermi level, a very small conductance is expected. However, for the spin-down channel, there is a very wide transmission peak starting at –0.2 eV and through the Fermi level, which gives rise to larger conductance than that of the spin-up channel. In addition, the spin-resolved current under bias V can be obtained by integrating the transmission spectra in the interval –eV/2 and eV/2, which is an intuitive explanation of why the spin-down current is much larger than the spin-up current. 

To quantitatively describe the spin injection efficiency, the spin filter efficiency (SFE) under the zero bias is defined as follows: SFE = [T_up_(E_F_) – T_down_(E_F_)]/[T_up_(E_F_) + T_down_(E_F_)]. Here, T_up_(E_F_) and T_down_(E_F_) are the transmission coefficients of the spin-up and spin-down electrons at the Fermi level, respectively. A positive or negative value of the SFE indicates conductance dominated by the spin-up or spin-down electrons. The T_up_(E_F_) and T_down_(E_F_) through the junction are about 2.20 ×10^-2^ and 0.27 G_0_ (G_0_ is the quantum conductance, and its value is e^2^/h), respectively; the SFE is predicted to be −84.7 %, indicating that spin-down electrons dominated. In fact, the real space local density of states (LDOS) of the spin-up and spin-down electrons are responsible for the different transport behaviors of two spin channels, as shown in Figure 5B. The delocalization of the LDOS for the spin-down electrons which can provides an effective transport channel at the Fermi level. While for the spin-up electrons, the LDOS do not distribute over all the scattering region, and then the transport channel is blocked at the Fermi level. Moreover, the DOS (Figure 6) also confirms that the transport channel of the spin-down electrons is more efficient. The spin-resolved DOS contributes to two C atoms that are nearest to the Au tip, a C−C bond adsorbed on the left electrode, and the Fe atom, respectively. As for the spin-up electrons, the magnitude of the DOS is very flat and tends to zero near the Fermi level. However, for the spin-down electrons, the magnitude of DOS is much larger than that of the spin-up electrons. We noticed that the DOS of the Fe has a significant contribution around the Fermi level, suggesting that the Fe atom plays a key role in the conduction of the spin-down electrons. The results of the real space LDOS and DOS are consistent, which explains the difference in conductance between the two spin channels.

At last, we examined the case of the Au tip bonding with Fe@C_28_. Pushing forward the STM tip about 1.30 Å, the Au–C bond is formed. Here, the molecule–electrodes coupling was stronger than that in the non-bonding case; therefore, the transmission peaks became higher and wider (Figure 6). The T_up_(E_F_) and T_down_(E_F_) are 9.17×10^−2^ and 0.53 G_0_, respectively. The SFE is predicted to be −70.4%, indicating that the conductance and SFE can be effectively modulated by the Au–C bond forming or breaking. All in all, our findings show that the Fe@C_28_ junction can be seen as a possible candidate for a high-efficiency spin filter, spin injector, and moderate rectifier under small bias voltage, which offers promising applications in nanoelectronic devices.

## 4. Conclusions

In summary, we investigated the Fe@C_28_ absorption properties on an Au(111) surface and transport properties by first principle DFT calculations and NEGF techniques. The free Fe@C_28_ has a localized 4.0-*µ*_B_ magnetic moment, and the cage tends to adsorb on the bridge sites of the Au(111) surface in the manner of a C–C bond. Our calculations show that the conductance-controllable Fe@C_28_ junction can act as a high-efficiency spin filter and as an effective spin injector under small bias voltage. A pure spin current can be generated by the Fe@C_28_ junction, which can be used as a spin source in a device, quantum computing, hard disk drive (HDD), etc. [8,9]. Moreover, when the magnetic moment of Fe@C_28_ is flipped by the external magnetic field, the spin signal is also reversed. Thus, the proposed Fe@C_28_ junction can also be used as a magnetic field sensor to detect the magnetic field [38]. All of the results suggest that Fe@C_28_ not only can be seen as a promising candidate for molecular spintronics but also will be helpful for designing novel fullerene-based spin filter devices.

## Figures and Tables

**Figure 1 nanomaterials-09-01068-f001:**
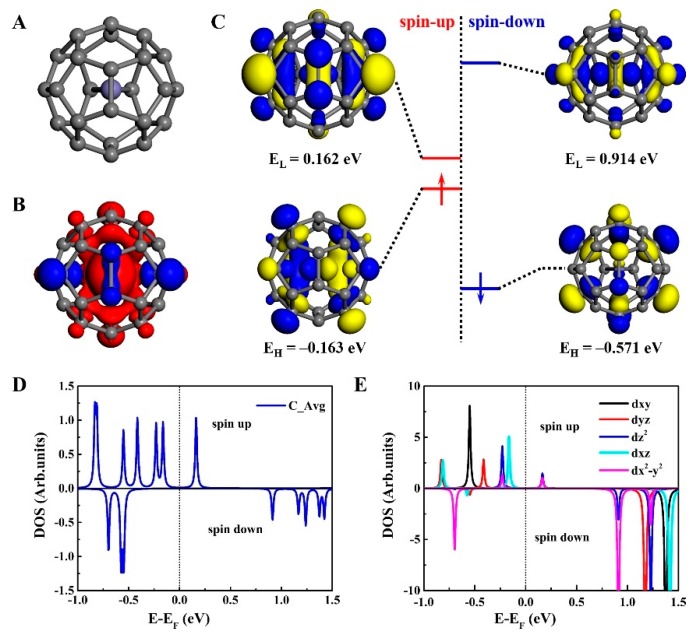
(**A**) Optimized structure of Fe@C_28_ fullerene with C2v symmetry. There are 10 inequivalent C atoms, 13 inequivalent C–C bonds, and 6 inequivalent rings, with the inequivalent atoms labeled by different colors. (**B**) Spin density of Fe@C_28_. (**C**) Spin-resolved frontier orbitals of Fe@C_28_. Isovalue is set to ±0.01 e/Å^3^. (**D**) Average spin-resolved density of states (DOS) projected to C28 of the Fe@C_28_ with the energy window from −1.0 to 1.5 eV. (**E**) Spin-resolved DOS projected to the five 3d orbitals of Fe atom.

**Figure 2 nanomaterials-09-01068-f002:**
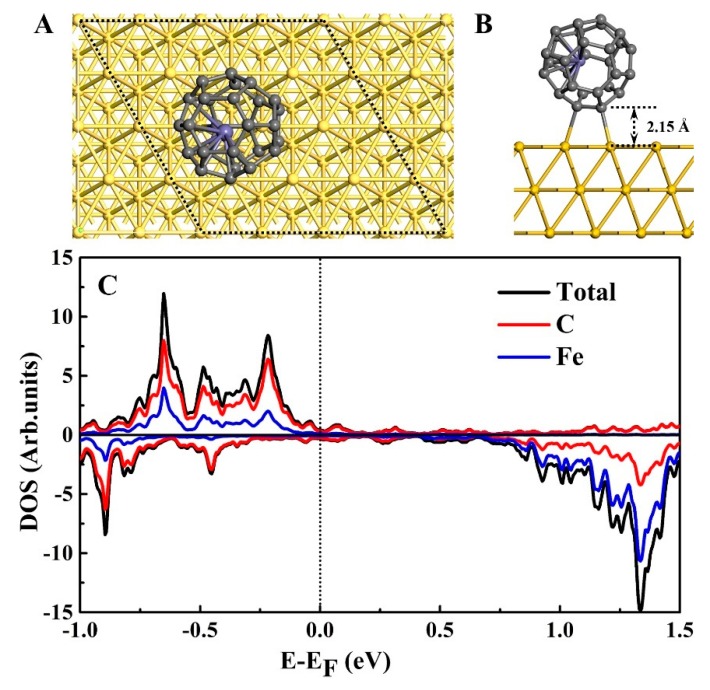
Most stable adsorption configuration. (**A**) Top view. (**B**) Side view. Fe@C_28_ is favored and adsorbed on the bridge site of the Au(111) surface via a C–C bond. (**C**) Density of states (DOS) of Fe@C_28_ after being adsorbed on the Au(111) surface within an energy window from –1.0 to 1.5 eV.

**Figure 3 nanomaterials-09-01068-f003:**
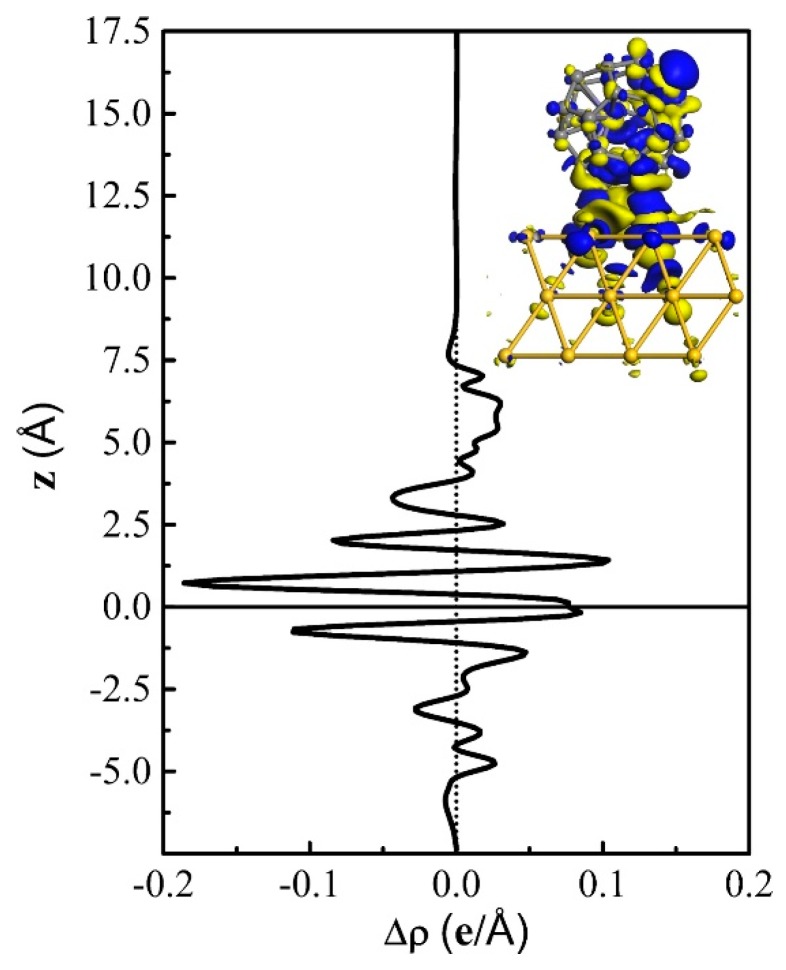
The *xy*-plane-averaged differential charge density is along the *z* direction. For simplicity, the *z* direction coordinates of the bottom Au layer are set to zero. Insert: differential charge density of the Fe@C_28_-Au adsorption configuration. The isovalue is set to ±0.008 e/Å^3^.

**Figure 4 nanomaterials-09-01068-f004:**
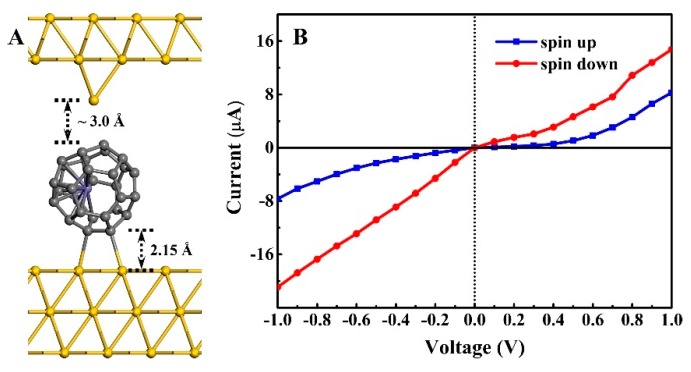
(**A**) Profile of the scanning tunneling microscopy (STM)-type Fe@C_28_ junction. The nearest Au–C distance is 3.12 Å. (**B**) Spin resolved current–voltage (I-V) curves within the voltage region from −1.0 to 1.0 V.

**Figure 5 nanomaterials-09-01068-f005:**
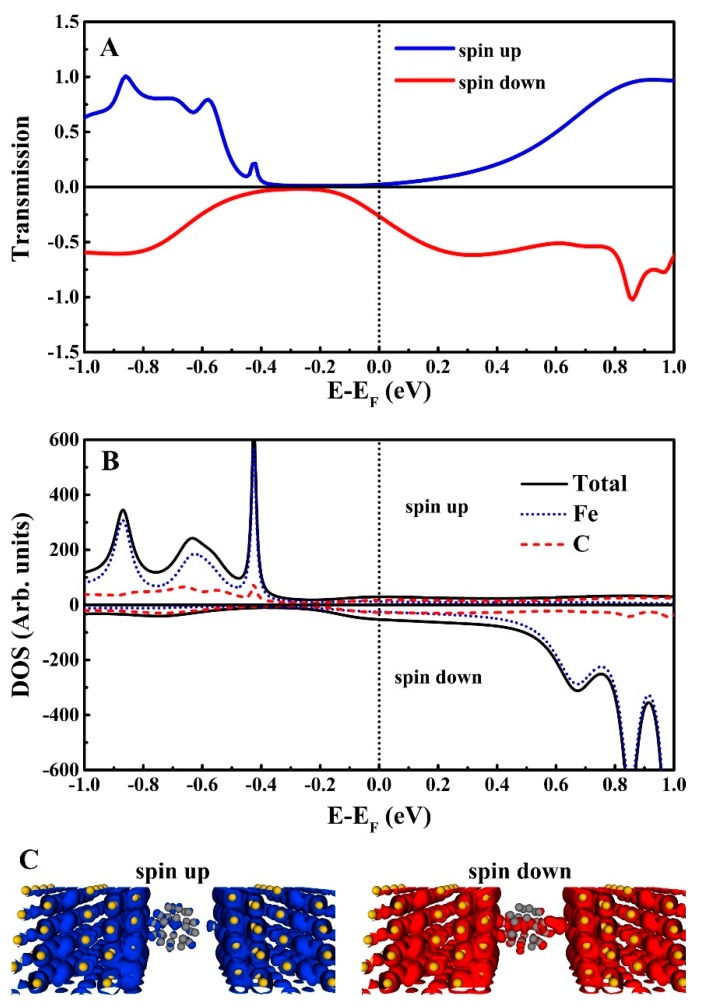
Transport properties of the Fe@C_28_ junction. Au tip and Fe@C_28_ are non-bonding. (**A**) Spin-resolved transmission spectra of the Fe@C_28_ junction. (**B**) Spin-resolved DOS projected on two C atoms that are nearest to the Au tip, a C–C bond adsorbed on the Au(111), and endohedral Fe, respectively. Black line: total DOS; dot: Fe; virtual line: C. (**C**) Profile of the spin-resolved local DOS at the Fermi level, which suggests that the spin-down electrons can offer an effective conductance channel at E_F_.

**Figure 6 nanomaterials-09-01068-f006:**
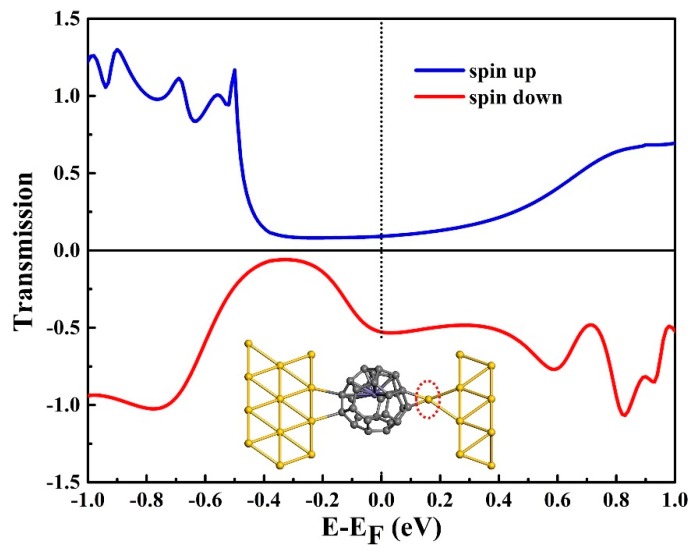
Spin-resolved transmission spectra of the molecular junction of Au tip bonding with Fe@C_28_. Insert: bonded Fe@C_28_ junction. The bonding area is denoted by the red dotted circle.

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
