# Peer review of "Endohedral Fullerene Fe@C28 Adsorbed on Au(111) Surface as a High-Efficiency Spin Filter: A Theoretical Study"

_nanomaterials, 2019, doi:10.3390/nano9081068_

Round 1
Reviewer 1 Report
The paper appears to be somewhat sound, but two concerns remain. First, DFT is only valid for ground state calculations and time-dependent DFT would be more reliable for non-equilibrium calculations. I assume that NEGF was used to find the transmission probabilities. I would like to see how the self-energy term was handled.
Author Response
We thank the reviewer for pointing this out. The theoretical effort and trend today is to move toward theories able to account for non-coherent and dissipative effects due to electron-phonon and electron-electron scattering mechanisms inside the conductor and for non-linear response, far from equilibrium, finite-bias transport. Along these directions, the two major research lines are Time-Dependent Density-Functional Theory (TDDFT) [Runge, E.K.U. Gross, W. Kohn]; and Non-Equilibrium Green's Function (NEGF) theory [Schwinger, Baym, Kadanoff, Keldysh]. Both NEGF and also TDDFT [G. Stefanucci, C.-O. Almbladh] are in principle correct frameworks to address the above objections.
The interaction between the device scattering region and the electrodes is accounted for by self-energies within the NEGF formalism. is related to the surface Green's function of a semi-infinite electrode: [Jeremy Taylor et al. Phys. Rev. B, 63, 245407 (2001); Mads Brandbyge, et al. Phys. Rev. B 65, 165401 (2002)]. VL (VR) is the interaction between the left electrode (right electrode), and central regions. The self-energy can be obtained based on the Recursion method [M. P. Lopez Sancho, et al. J. Phys. F: Metal Physics, 15, 851 (1985)].
Reviewer 2 Report
The authors present a theoretical study of the electronic structure of a Fe@C28 molecule followed by a study of its adsorption on an Au(111) surface. Finally, they study the transport through an Au(111)surface/Fe@C28/Au(tip) junction. Their approach is based on DFT/Green function calculations using the SIESTA and NanoDCal software packages. Two interesting results in this paper include (1) the demonstration that the Fe@C28 molecule has a magnetic moment of about 4 Bohr magneton, and (2) the Au(111)surface/Fe@C28/Au(tip) can act as an efficient spin filter at small applied bias because transport through the junction is mostly due to spin-down electrons.
These results are interesting from a fundamental physics viewpoint. Practical realizations of these spin injectors in much larger circuits requiring many of these junctions could be problematic. The junctions could maybe used in parallel for detection of very small magnetic field. The paper is suitable for publication if the authors add more potential realistic applications of the proposed junction.
Author Response
We thank the reviewer for improving our manuscript. In the revision, we have add “Our results show that the Fe@C28 junction is not only a highly efficient spin filter in equilibrium, but also an effective spin injector under small bias.” and “such as spin-filter with large spin injection coefficients and molecular memories”.

Reviewer 3 Report
The scientific content of the ms is interesting and this work thus deserves – according to my opinion – acceptance and publication in NANOMATERIALS. I am sure that the paper will attract the interest of scientists working in the broad field of molecular spintronics, and especially in the area of the chemistry and properties of magnetic endohedral fullerenes. Also, I do believe that the article will receive a respectable number of citations in the future. Salient features of this work (which is theoretical)- that support my proposal for acceptance – are: (a) The study completes a gap in the literature because the spin transport properties of the magnetic transition metal@C28 fullerenes were not studied in the past. (b) The free Fe@C28 species has a localized magnetic moment of 4.0 μΒ and the cage tends to be adsorbed on the bridge sites of the Au(111) surface across the C-C bonds. (c) The spin resolved transport calculations show that the Fe@C28 junction can be modulated. Overall, the study predicts that Fe@C28 is a promising candidate for molecular spintronics. The ms is well organized and the quality of figures is good. The references list covers the topic under study satisfactorily.
Based on the above mentioned, I am glad to propose acceptance of this work in NANOMATERIALS. I do not have scientific points to raise. Minor comments to be taken into account by the authors:
(1) “Title”: The theoretical nature of the present study should be emphasized.
(2) “Abstract”: SFE should be defined.
(3) The English of the ms should be improved.
(4) I would welcome the addition of 2-3 sentences, briefly explaining equations (1) and (2). This should be helpful for the non-familiar readers.
Author Response
We thank the reviewer for improving our manuscript.
(1) The title is modified and add “a theoretical study”
(2) SFE is defined in the abstract.
(3) We have fixed a lot of typos and grammatical errors, and have rewritten some ambiguous statements, please see the “Mainly Change List”
(4) We have added “Iσ(V) is the current through the device under the external bias voltage V” and “indicating the rate at which electrons transmit from the left to the right electrodes by propagating through the device” in the section of Materials and Methods.

Round 2
Reviewer 1 Report
I am satisfied with the response. The English still needs some improvement.
Author Response
Response 1: We thank the reviewer for improving our manuscript.
[1] We have rewritten some ambiguous statements. For clarity, we also delete some meaning duplicate sentences.
[2] In the revision, we have added two applications of the proposed junction: “A pure spin current can be generated by the Fe@C28 junction, which can be used as a spin source in a device, quantum computing and hard disk drive (HDD) etc. [8,9]. Moreover, when the magnetic moment of Fe@C28 is flipped by external magnetic field, the spin signal is also reversed. Thus, the proposed Fe@C28 junction can also be used as a magnetic field sensor to detect the magnetic field [38].”and cite a new reference labelled with [38].
[3] We have fixed some typos and grammatical errors in manuscript e.g. all the “electron structure” are changed to “electronic structure”.
